# YFS resummation for future lepton-lepton colliders in SHERPA

**Frank Krauss[1], Alan Price[2] and Marek Schönherr[1]**

**1** Institute for Particle Physics Phenomenology, Durham University, United Kingdom
**2** University of Siegen, Department of Physics, 57068 Siegen, Germany

## Abstract

We present an implementation of the Yennie–Frautschi–Suura (YFS) scheme for the all-orders resummation of logarithms from the emission of soft real and virtual photons in processes that are critical for future lepton colliders. They include, in particular, $e^-e^+ \to f\bar{f}$ and $e^-e^+ \to W^-W^+$, where we validate the results of our implementation, improved with fixed-order corrections, with those obtained from the most precise calculations. We also show, for the first time, results for the Higgs-Strahlungs process, $e^-e^+ \to ZH$, in YFS resummation including fixed-order improvements up to order $\alpha^3 L^3$.

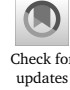
## 1 Introduction

With an unprecedented amount of data gathered, the Large Hadron Collider (LHC) continues to provide a rich environment, that allows us to further test and scrutinise our knowledge of

fundamental particles and their interactions with matter. The discovery of the Higgs boson by the ATLAS [1] and CMS [2] collaborations provided conclusive validation of spontaneously broken gauge theories as the construction principle underpinning our understanding of Nature, with the Standard Model (SM) of Particle Physics as its manifest realization. To date, there has not been any direct discovery of physics beyond the SM at the LHC, but there are observations that cannot be fully explained within its framework: for example, the observation of non-zero neutrino masses, dark matter and energy, and the anti-matter matter asymmetry. Being intimately tied to the generation of fundamental masses and, through them, CP violation as encoded in the CKM matrix, the Higgs-Boson may provide key inputs to answer some of these questions.

While the LHC, and potential hadron collider successors, may yet provide a deeper understanding to the nature of the Higgs-Boson, a future lepton-lepton collider or "Higgs-Factory" could provide unprecedented measurements of the electroweak nature of the SM and thus provide an alternative experimental avenue for the community to explore [3]. This is also stated in the European Strategy update from 2020 [4]: "*An electron-positron Higgs-factory is the highest-priority next collider*". Due to their beams composed of elementary particles, unlike a hadron machine, a lepton-lepton collider has a very clean initial state, which facilitates measurements of electroweak pseudo-observables (EWPO) [5] with unprecedented precision. One of these collider designs is the Future Circular Collider, a post-LHC particle accelerator, that initially would be operate in an $e^+e^-$ mode (FCC-ee) [6,7] before being repurposed to run as a high energy hadron-hadron collider (FCC-hh). For the initial leptonic operations four centre-of-mass energies have been proposed: After a first four-year stage, where the collider would run at the Z-pole and collect a projected 150 ab$^{-1}$ of data, corresponding to the production of a staggering $10^{12}$ $Z$ bosons [6], the energy would be increased for two years to $\sqrt{s} = 161$ GeV, the $W^+W^-$ production threshold, with a projected integrated luminosity of 12 ab$^{-1}$ or $10^8$ $W^+W^-$ pairs. After these two first phases, a three-year run at $\sqrt{s} = 240$ GeV would turn the collider into a Higgs factory, with copious production of the Higgs boson through the Higgsstrahlung process $e^+e^- \to ZH$ and 1 million $ZH$ events at an integrated luminosity of 5 ab$^{-1}$. While this energy does not correspond to the maximum cross section for Higgsstrahlung, it would maximize the event per unit time due to the colliders luminosity profile. This run will produce 1 million ZH events. The last phase of operations would be at the $t\bar{t}$ production threshold, with a multipoint scan around the threshold range $\sqrt{s} = 345 - 365$ GeV and an integrated luminosity of 1.5 ab$^{-1}$, resulting in a million top-pair production events. As an alternative option for an $e^+e^-$ collider, the Circular Electron Positron Collider (CEPC) [8,9], is projected as a China-based Higgs-factory with a circumference of 80 km. Similar to the FCC-ee it is designed to operate at different centre-of-mass energies during subsequent operation phases, namely at 91.2 GeV as a $Z$-factory, producing close to $10^{12}$ $Z$ bosons, at 160 GeV for of the production of $10^8$ $W$ bosons at threshold, and at 240 GeV as a Higgs-factory, resulting in $10^6$ Higgs bosons. Like the FCC-ee, there is also the possibility for the CEPC to be transformed into a hadron-hadron collider at later stages.

While circular colliders can reach large integrated luminosities, they are rather restricted in their energy reach due to QED Bremsstrahlung effects. This is not a primary concern for linear colliders which can thus reach energies in the multi-TeV range and provide polarised beams further amplifying their physics potential, but they will operate at significantly reduced luminosities. The Compact Linear Collider (CLIC) [10–12] has been proposed as a multi-TeV linear $e^+e^-$ machine, to be built at CERN. It would operate in three stages, with centre-of-mass energies at 380 GeV, 1.5 TeV, and 3 TeV. In the first stage, about 160,000 Higgs bosons will be produced through Higgsstrahlung and $WW$ fusion while during operations in the TeV range, it is expected to produce millions of Higgs-bosons. The final $e^+e^-$ collider being under current consideration is the International Linear Collider (ILC) [13–16] in Japan. Being smaller than

Table 1: The current systematic and statistical uncertainties on QED sensitive observables, with terms in {...} denoting the contributions to QED alone. The FCC-ee uncertainty estimates have been taken from [21], the overall table has been reproduced from [22]

| Observable | Where from | Current (LEP) | FCC (stat.) | FCC (syst.) | $\frac{\text{Now}}{\text{FCC}}$ |
|---|---|---|---|---|---|
| $M_Z$ [MeV] | $Z$ linesh. [17] | $91187.5 \pm 2.1\{0.3\}$ | $0.005$ | $0.1$ | $3$ |
| $\Gamma_Z$ [MeV] | $Z$ linesh. [17] | $2495.2 \pm 2.1\{0.2\}$ | $0.008$ | $0.1$ | $2$ |
| $R_l^Z = \Gamma_h/\Gamma_l$ | $\sigma(M_Z)$ [18] | $20.767 \pm 0.025\{0.012\}$ | $6 \cdot 10^{-5}$ | $1 \cdot 10^{-3}$ | $12$ |
| $\sigma_{\text{had}}^0$ [nb] | $\sigma_{\text{had}}^0$ [17] | $41.541 \pm 0.037\{0.025\}$ | $0.1 \cdot 10^{-3}$ | $4 \cdot 10^{-3}$ | $6$ |
| $N_\nu$ | $\sigma(M_Z)$ [17] | $2.984 \pm 0.008\{0.006\}$ | $5 \cdot 10^{-6}$ | $1 \cdot 10^{-3}$ | $6$ |
| $N_\nu$ | $Z\gamma$ [19] | $2.69 \pm 0.15\{0.06\}$ | $0.8 \cdot 10^{-3}$ | $< 10^{-3}$ | $60$ |
| $\sin^2\theta_W^{eff} \times 10^5$ | $A_{FB}^{lept.}$ [18] | $23099 \pm 53\{28\}$ | $0.3$ | $0.5$ | $55$ |
| $\sin^2\theta_W^{eff} \times 10^5$ | $\langle\mathcal{P}_\tau\rangle, A_{FB}^{pol,\tau}$ [17] | $23159 \pm 41\{12\}$ | $0.6$ | $< 0.6$ | $20$ |
| $M_W$ [MeV] | ADLO [20] | $80376 \pm 33\{6\}$ | $0.5$ | $0.3$ | $12$ |
| $A_{FB,\mu}^{M_Z \pm 3.5\,\text{GeV}}$ | $\frac{d\sigma}{d\cos\theta}$ [17] | $\pm 0.020\{0.001\}$ | $1.0 \cdot 10^{-5}$ | $0.3 \cdot 10^{-5}$ | $100$ |

CLIC, it is expected to run at multiple energies from the Z-pole up to 500 GeV, with a possible later upgrade to 1 TeV. Dedicated runs at 91 GeV and 160 GeV will focus on the $Z$ boson in $e^+e^- \to f\bar{f}$ and the production of $W$ pairs at threshold, $e^+e^- \to W^+W^-$, respectively. The run at 250 GeV will allow precision studies of the Higgs-boson through $e^+e^- \to ZH$ with a projected integrated luminosity of 250 fb$^{-1}$. Final runs at the $t\bar{t}$ threshold and at 500 GeV could be further supplemented through machine upgrades that would allow to probe couplings of the Higgs boson to the top quark and to itself, with runs at 1 TeV.

To take full advantage of these highly precise machines, the corresponding theory uncertainties have to be smaller or, at least, match the size of the experimental ones, *cf.* Table 1 for some examples. There are two main types of theory uncertainties that need to be addressed by the community for the successful analysis of the experimental results: parametric uncertainties, which reflect our limited knowledge of the fundamental SM input parameters, and uncertainties due to missing higher-order terms in our perturbative calculations. A large source of intrinsic theoretical uncertainties are corrections due to QED radiation, where improvements by factors of 2-100, depending on the observable, are mandatory to reduce the QED uncertainties to an acceptable level. This demand reflects the simple fact that QED effects of the order 0.1%, that could be safely ignored ignored at LEP, turn into limiting factors in the full analysis of experimental results at future Higgs-Factories. For example, from a precise measurement of the total hadronic cross section of $e^+e^- \to f\bar{f}$ near the Z-pole, the mass and width of the Z boson can measured with an uncertainty of the order of 0.1 MeV. This translates into the corresponding QED uncertainty that will have to be reduced to $\delta_{QED} \leq 0.03$ MeV [23], mandating the inclusion of corrections up to $\mathcal{O}(\alpha^4 L^4)$ with $L = \log(s/m_f^2)$.

This paper reports first steps towards achieving the necessary theoretical precision within the framework of the SHERPA event generator [24–26], with a particular focus on the description of photon radiation off the incoming leptons. In SHERPA, as in many other Monte Carlo event generators [27–29], this has so far been included through the structure function approach [30] which resums the large logarithms associated to multiple (collinear) photon emission through the Dokshitser–Gribov–Lipatov–Altarelli–Parisi (DGLAP) equations [31–34]. While SHERPA implements the well-known leading-order accuracy [35–37], in recent years the structure functions have been extended to next-to-leading logarithmic accuracy [38, 39]. At fixed order, sub-leading logarithmic corrections to $e^+e^- \to \gamma^*/Z$ have been calculated up to $\mathcal{O}(\alpha^6 L^6)$ [40–42].

However, instead of further improving the structure function approach implementation in

SHERPA, we will here calculate the emission of soft photons and resum the associated logarithms in the Yennie–Frautschi–Suura formulation (YFS) [43]. In YFS, and in contrast to the structure function approach, photon emissions are treated in a fully differential form, explicitly creating the photons in an exact treatment of the emission phase space and the recoil. We will built on previous experiences with the YFS method, applied to final state radiation in decays of unstable particles in the SHERPA's PHOTONS module [44], which has recently been supplemented with the inclusion of weak corrections and the exact treatment of second-order QED effects in the decays of $Z$ and Higgs bosons [45]. We will present a process-independent implementation of the YFS method to initial state radiation (ISR), resumming the associated soft logarithms to all orders, and we will add the interplay with QED emissions in final-state radiation (FSR). We will further add explicit higher-order QED and weak corrections for certain processes, in particular for $e^+e^- \to f\bar{f}$, $e^+e^- \to W^+W^-$, and for $e^+e^- \to HZ \to H\ell^+\ell^-$ which will pave the way for the systematic inclusion of such matrix element corrections into the resummation framework.

In Section 2 we will summarise the YFS approach to the description of photon emissions to all orders and the corresponding resummation of the associated logarithms. This will be followed in Section 3 by the careful validation of our results, comparing them with equivalent results obtained from KKMC [46, 47] for $e^+e^- \to f\bar{f}$ and KORALW/YFSWW [48] for $e^+e^- \to W^+W^-$. We will quickly turn to a comparison with results obtained from a structure function approach for the description of QED initial state radiation in Section 4. In Section 5 we will show and discuss our results for the Higgs–Strahlungs process $e^+e^- \to HZ \to H\ell^+\ell^-$ in YFS resummation, which is of course of paramount interest for the success of experiments at future $e^+e^-$ colliders. We will summarise our work in Section 6, where we will also detail some future steps.

## 2 Theory

The Yennie-Frautschi-Suura (YFS) formalism [43] provides a robust method for resumming, to all orders, the potentially large logarithms that are associated with the emission of real and virtual photons in the soft limit. YFS systematically incorporates further improvements of the theoretical accuracy of the resummation through the inclusion of exact fixed-order expressions. In contrast to the structure function method, the YFS approach *explicitly* generates resolved photons, with a resolution criterion given by an energy and angle cut-off. In so doing the full kinematic structure of scattering events is reconstructed which leads to a straightforward implementation of the YFS method as both cross section calculator and event generator. We will detail the relevant algorithms in App. B and focus here on a discussion of the theory background only.

Summing over all real and virtual photon emissions, the total cross section for a $2 \to N$ scattering process with kinematics $p_1 + p_2 \to p_3 + p_4 \cdots + p_{N+2}$ is given by,

$$d\sigma = \sum_{n_\gamma=0}^\infty \frac{1}{n_\gamma!} d\Phi_Q \left[ \prod_{i=1}^{n_\gamma} d\Phi_i^\gamma \right] (2\pi)^4 \delta^4 \left( \sum_{i=1}^2 p_i - \sum_{j=3}^{N+2} q_j - \sum_{k=1}^{n_\gamma} k_k \right) \left| \sum_{\bar{n}_\gamma=0}^\infty \mathcal{M}_{n_\gamma}^{\bar{n}_\gamma + \frac{1}{2} n_\gamma} \right|^2, \quad (1)$$

where the outgoing momenta $q_j$ emerge from the original $p_j$ after the effect of the real photon emissions has been taken into account. Here, $d\Phi_Q$ denotes the modified final state phase space element, the $d\Phi_i^\gamma$ are the phase space elements spanned by the $n_\gamma$ real photon momenta $k_i$ emitted off the leading order configuration. Similarly, $\bar{n}_\gamma$ counts the number of virtual photons added to it. The indices $i$ and $j$ matrix elements $\mathcal{M}_i^j$ indicate the number of real photons, $i$, and the overall additional orders in $\alpha$, $j$, relative to the Born configuration. Consequently

the Born matrix element for the core process without any additional real or virtual photons is designated as $\mathcal{M}_0^0$. This expression includes photon emissions to all orders, but it is obviously an unrealistic ambition to be able to calculate all its contributing terms in Eq. (1); instead we may only calculate the first few terms in the perturbative series, until a target accuracy is reached.

## Exponentiation of virtual photon contributions

To analyse the structure of the resummation and its fixed-order improvements, consider the case of a single virtual photon. In the soft limit, the matrix element factorises as

$$\mathcal{M}_0^1 = \alpha B M_0^0 + M_0^1 . \tag{2}$$

Therein $\alpha$ is the QED coupling constant, $B$ is an integrated off-shell eikonal encoding the universal soft-photon limit, see App. A, and $\mathcal{M}_0^0 = M_0^0$ is the leading order matrix element. Then, the finite remainder $M_0^1$ is the infrared-subtracted matrix element including one virtual photon. YFS showed that the insertion of further virtual photons leads to a power series; for up to two virtual photons we have

$$\begin{aligned}
\mathcal{M}_0^0 &= M_0^0 , \\
\mathcal{M}_0^1 &= M_0^1 + \alpha B M_0^0 , \\
\mathcal{M}_0^2 &= M_0^2 + \alpha B M_0^1 + \frac{(\alpha B)^2}{2!} M_0^0 ,
\end{aligned} \tag{3}$$

where $M_0^2$ is now the infrared finite remainder of the matrix element containing two virtual photons. This generalises to any number $\bar{n}_\gamma$ of virtual photons,

$$\mathcal{M}_0^{\bar{n}_\gamma} = \sum_{r=0}^{\bar{n}_\gamma} M_0^{\bar{n}_\gamma - r} \frac{(\alpha B)^r}{r!} , \tag{4}$$

and resumming all virtual photon emissions therefore gives

$$\sum_{\bar{n}_\gamma=0}^{\infty} \mathcal{M}_0^{\bar{n}_\gamma} = \exp(\alpha B) \sum_{\bar{n}_\gamma=0}^{\infty} M_0^{\bar{n}_\gamma} . \tag{5}$$

Due to the abelian nature of QED and the absence of collinear singularities in the soft-photon limit, this can be further generalised to squared matrix elements that include any number of additional real photon emissions, such that

$$\left| \sum_{\bar{n}_\gamma=0}^{\infty} \mathcal{M}_{n_\gamma}^{\bar{n}_\gamma + \frac{1}{2} n_\gamma} \right|^2 = \exp(2\alpha B) \left| \sum_{\bar{n}_\gamma} M_{n_\gamma}^{\bar{n}_\gamma + \frac{1}{2} n_\gamma} \right|^2 . \tag{6}$$

By construction, $M_{n_\gamma}^{\bar{n}_\gamma + \frac{1}{2} n_\gamma}$ is completely free of soft singularities due to virtual photons but it will still contain those due to real photons.

## Exponentiation of real photon contributions

For real photon emissions, the factorization occurs at the level of squared matrix elements. For the case of a single real photon this can be expressed as,

$$\frac{1}{2(2\pi)^3} \left| \sum_{\bar{n}_\gamma=0}^{\infty} M_1^{\bar{n}_\gamma + \frac{1}{2}} \right|^2 = \tilde{S}(k) \left| \sum_{\bar{n}_\gamma=0}^{\infty} M_0^{\bar{n}_\gamma} \right|^2 + \sum_{\bar{n}_\gamma=0}^{\infty} \tilde{\beta}_1^{\bar{n}_\gamma + 1}(k) . \tag{7}$$

In this expression, all the (residual real emission) singularities are contained within the eikonal, $\tilde{S}(k)$, defined in Eq. (B.4). The $\tilde{\beta}_{n_\gamma}^{\bar{n}_\gamma + n_\gamma}$ are the infrared-finite squared matrix elements. They correspond to the Born level process plus emissions of $n_\gamma$ real and $\bar{n}_\gamma$ virtual photons at order $n_\gamma + \bar{n}_\gamma$ in the QED coupling $\alpha$. For convenience we introduce the following notation,

$$\tilde{\beta}_{n_\gamma} = \sum_{\bar{n}_\gamma = 0}^{\infty} \tilde{\beta}_{n_\gamma}^{\bar{n}_\gamma + n_\gamma}\,. \tag{8}$$

Extracting all real-emission soft photon divergences through eikonal factors, the squared matrix element for any $n_\gamma$ real emissions, summed over all possible virtual photon corrections, can be written as

$$
\left(\frac{1}{2(2\pi)^3}\right)^{n_\gamma} \left|\sum_{\bar{n}_\gamma = 0}^{\infty} M_{n_\gamma}^{\bar{n}_\gamma + \frac{1}{2}n_\gamma}\right|^2
$$

$$
= \tilde{\beta}_0 \prod_{i=1}^{n_\gamma}\left[\tilde{S}(k_i)\right] + \sum_{i=1}^{n_\gamma}\left[\frac{\tilde{\beta}_1(k_i)}{\tilde{S}(k_i)}\right]\prod_{j=1}^{n_\gamma}\left[\tilde{S}(k_j)\right] + \sum_{\substack{i,j=1\\i<j}}^{n_\gamma}\left[\frac{\tilde{\beta}_2(k_i,k_j)}{\tilde{S}(k_i)\tilde{S}(k_j)}\right]\prod_{l=1}^{n_\gamma}\left[\tilde{S}(k_l)\right] + \ldots
$$

$$
+ \tilde{\beta}_{n_\gamma - 1}(k_1,\ldots,k_{i-1},k_{i+1},\ldots,k_{n_\gamma})\sum_{i=1}^{n_\gamma}\tilde{S}(k_i) + \tilde{\beta}_{n_\gamma}(k_1,\ldots,k_{n_\gamma})\,. \tag{9}
$$

Within this expression, all $\tilde{\beta}_i$ are free from all infrared divergences due to either real or virtual photon emissions.

The first term in Eq. (9), $\tilde{\beta}_0$, contains all virtual photon corrections to the Born matrix element and approximates the emission of $n_\gamma$ real photons through the eikonals $\tilde{S}(k)$. The second term corrects the eikonal approximation for one photon at a time to the exact single photon emission matrix element, including all virtual corrections to it. Similarly, the next term corrects the coherent emission of two real photons to the exact expression including all virtual corrections, and so on.

In order to recombine all terms into an expression for the inclusive cross section and facilitate the cancellation of all infrared singularities it is useful to define an unresolved region $\Omega$ in which the kinematic impact of any real photon emission is unimportant. Integrating over this unresolved real emission phase space gives the integrated on-shell eikonal $\tilde{B}$,

$$2\alpha\tilde{B}(\Omega) = \int \frac{\mathrm{d}^3 k}{k^0}\tilde{S}(k)\left[1 - \Theta(k,\Omega)\right]\,, \tag{10}$$

which contains all infrared poles due to real soft photon emission.[1] Substituting this expression back into Eq. (1), the contributions originating from $\tilde{B}$ for all $n_\gamma$ photons again exponentiate. This gives

$$\mathrm{d}\sigma = \sum_{n_\gamma = 0}^{\infty} \frac{e^{Y(\Omega)}}{n_\gamma!}\mathrm{d}\Phi_Q\left[\prod_{i=1}^{n_\gamma}\mathrm{d}\Phi_i^\gamma\,\tilde{S}(k_i)\,\Theta(k_i,\Omega)\right]\left(\tilde{\beta}_0 + \sum_{j=1}^{n_\gamma}\frac{\tilde{\beta}_1(k_j)}{\tilde{S}(k_j)} + \sum_{\substack{j,k=1\\j<k}}^{n_\gamma}\frac{\tilde{\beta}_2(k_j,k_k)}{\tilde{S}(k_j)\tilde{S}(k_k)} + \cdots\right), \tag{11}$$

with the YFS form-factor

$$Y(\Omega) = 2\alpha\left[B + \tilde{B}(\Omega)\right]\,. \tag{12}$$

Therein, all infrared singularities originating from real and virtual soft photon emission, contained in $\tilde{B}$ and $B$ respectively, cancel, leaving a finite remainder. An explicit expression for the form-factor can be found in App. A.

---

[1] $\Theta(k,\Omega) = 1$ if the photon $k$ does not reside in $\Omega$ and zero otherwise.

Finally, let us comment on a technical complication in Eq. (11), related to the question on how to evaluate matrix elements for the emission of a fixed number of photons when the event itself contains many more soft photons. For example, the $\tilde{\beta}_1$ terms are defined in the $(N+1)$-particle phase space while the full event populates an $(N+n_\gamma)$-particle phase space. This necessitates the projection of the momenta of the latter onto a phase space with lower dimension [46] and reflects the fact that the subtraction, and, consequently, the calculation of the $\tilde{\beta}_i$, proceeds at the end-point where the momenta of the emitted photons vanish. In our example of the evaluation of the matrix correction for one real photon, $\tilde{\beta}_1$, the phase space projection must satisfy four-momentum conservation:

$$\sum_{i=1}^{2} p_i = \sum_{j=3}^{N+2} q_j + \sum_{k=1}^{n_\gamma} k_k \;\longrightarrow\; \sum_{i=1}^{2} \mathcal{R} p_i = \sum_{j=3}^{N+2} \mathcal{R} q_j + k_1 \,. \tag{13}$$

In this reduction step, there is some freedom in how the projection $\mathcal{R}$ is chosen, exploiting the Lorenz invariance of the matrix elements and the phase space. Different choices will lead to different resulting, additional Jacobeans, and it is advantageous to employ such mappings $q \rightarrow \mathcal{R}q$, where the Jacobean us unitary. We detail how we generate the photon momentum in App. B.

## Collinear logarithms: EEX vs. CEEX exponentiation

While fixed-order perturbative calculations truncate the expansion at some predefined order, for example at $\mathcal{O}(\alpha^n)$, resummed calculations include certain classes of logarithms times coupling constants to all orders. The YFS scheme resums, to all orders, the logarithms emerging from soft real and virtual photon emissions, but misses the collinear logarithms in the exponentiation, which, formally speaking are of the same size. Instead, the inclusion of these logarithms is delegated to the infrared, or, better, soft–finite hard remainders $\tilde{\beta}$. In principle, of course, these corrections could be included to all orders, but in practice this series is truncated at a given order $\mathcal{O}(\alpha^n L^m)$, with $L$ the collinear logarithm $L = \log(s/m_i^2)$.

There are two approaches for inclusion of these higher order effects. The first, known as exclusive exponentiation (EEX), closely follows the logic in the original YFS paper by constructing the $\tilde{\beta}_i^j$ using analytical differential distributions from the corresponding Feynman diagrams. These expressions are given in terms of products of fours vectors – at low orders usually Mandelstam variables or similar quantities. They have the advantage that they are relatively straightforward to implement and their behaviour in certain limits, like the soft or collinear limits, can be easily checked for the correct behaviour. Independently constructing terms corresponding to initial and final state radiation (ISR and FSR) contributions facilitates the simple automation and implementation of the ISR contributions for lepton ($e^+e^-$ or $\mu^+\mu^-$) colliders. Due to the possibly complex final states such a simple automation is not feasible for the FSR parts, and the corresponding contributions have to be included on a case–by–case basis. The EEX corrections that we have included have been calculated in [30, 49–54] and the explicit expressions for the IR subtracted terms can be found in [55].

In the second approach, known as coherent exclusive exponentiation (CEEX) [55], the soft–finite $\tilde{\beta}_i^j$ are constructed at the amplitude level and therefore the subtraction is performed before squaring the amplitudes. In turn, they will automatically include all ISR, FSR and ISR-FSR interference effects as well as a consistent treatment of any spin effects. The downside of this theoretically superior approach is the drastically increased difficulty compared to EEX, which presents a veritable obstacle to its straightforward and transparent automation. As a consequence, CEEX so far has only been implemented for $e^+e^- \rightarrow f\bar{f}$, and an algorithm underpinning an $e^+e^- \rightarrow W^+W^-$ implementation has been described in [56].

Table 2: The explicit beta terms that have been implemented in SHERPA's EEX. The ISR corrections are universal and are applied to any lepton-collider process. For the FSR the corrections are only applied to the $f\bar{f}$ dipoles from decays of neutral bosons. The corrections to the decay of charged bosons have been implemented in the PHOTONS module [44,45] and the two methods will be combined in future work. The reference column provides the equation number from [55] where the explicit form of the corrections are given.

| Order | ISR Corrections | FSR Corrections | Reference |
|---|---|---|---|
| $\tilde{\beta}_0^0$ | Born | Born | |
| $\tilde{\beta}_1^1 + \tilde{\beta}_0^1$ | $\mathcal{O}(\alpha, \alpha L)$ | $\mathcal{O}(\alpha, \alpha L)$ | Eq. (13) |
| $\tilde{\beta}_2^2 + \tilde{\beta}_1^2 + \tilde{\beta}_0^2$ | $\mathcal{O}(\alpha^2 L^2)$ | $\mathcal{O}(\alpha^2 L)$ | Eq. (18,19) |
| $\tilde{\beta}_3^3 + \tilde{\beta}_2^3 + \tilde{\beta}_1^3$ | $\mathcal{O}(\alpha^3 L^3)$ | — | Eq. (26) |

**Infrared Boundary for FSR and ISR**

To combine the effects of soft photon emissions from ISR and FSR, it is important to guarantee the absence of double-counting, either positive or negative. Following the literature and other implementations, e.g. in HERWIG [57], in the EEX approach implemented in SHERPA this is achieved by analysing the singular domains in each case. They are given by rejecting photons with an energy of $k^0$ in the c.m. frame of the emitting dipole creating the eikonal as

$$\Omega = \begin{cases} k_I^0 < \epsilon_I \cdot \dfrac{\sqrt{s_Q}}{2}, & \text{for} \quad \Omega_I, \\[2ex] k_F^0 < \epsilon_F \cdot \dfrac{\sqrt{s_Q}}{2}\left(1 + \dfrac{2\sum_i k_i \cdot Q}{s_Q}\right), & \text{for} \quad \Omega_F, \end{cases} \tag{14}$$

where $s_Q = (q_1 + q_2)^2$, the c.m.-energy squared after initial photons have been radiated off. These two regions need to be consistently combined, for example [47] by choosing $\epsilon_F$ small enough such that the FSR domain lies with the ISR domain. This requires to also remove any FSR photon that resides in the overlap region $\delta\Omega = \Omega_I \setminus \Omega_F$. This removal process does not come for free. As we are in essence redefining the IR domain for the FSR photons, the corresponding phase space integration has changed and the contribution of the double-counted region to the overall exponentiated form factor has to be removed. For each dipole constituting an FSR eikonal with charges $Q_f$, this removal amounts to multiplication with a factor

$$\begin{aligned} W_{\text{FSR/ISR removal}} = \ & \exp\Bigg[ -2\alpha Q_f^2 \big[\tilde{B}(\Omega_I; \bar{q}_1, \bar{q}_2) - \tilde{B}(\Omega_F; \bar{q}_1, \bar{q}_2)\big] \\ & + 2\alpha Q_f^2 \big[\tilde{B}(\Omega_I; q_1, q_2) - \tilde{B}(\Omega_F; q_1, q_2)\big]\Bigg], \end{aligned} \tag{15}$$

where $q_{1,2}$ are the final state momenta after the photon emission and $\bar{q}_{1,2}$ are scaled momenta such that $\bar{q}_i^2 = m_i^2 \frac{s_Q}{s}$. The second term in the exponential is used to remove $\tilde{B}(\Omega_F)$ from the original YFS form factor as $\Omega_I$ now includes $\Omega_F$.

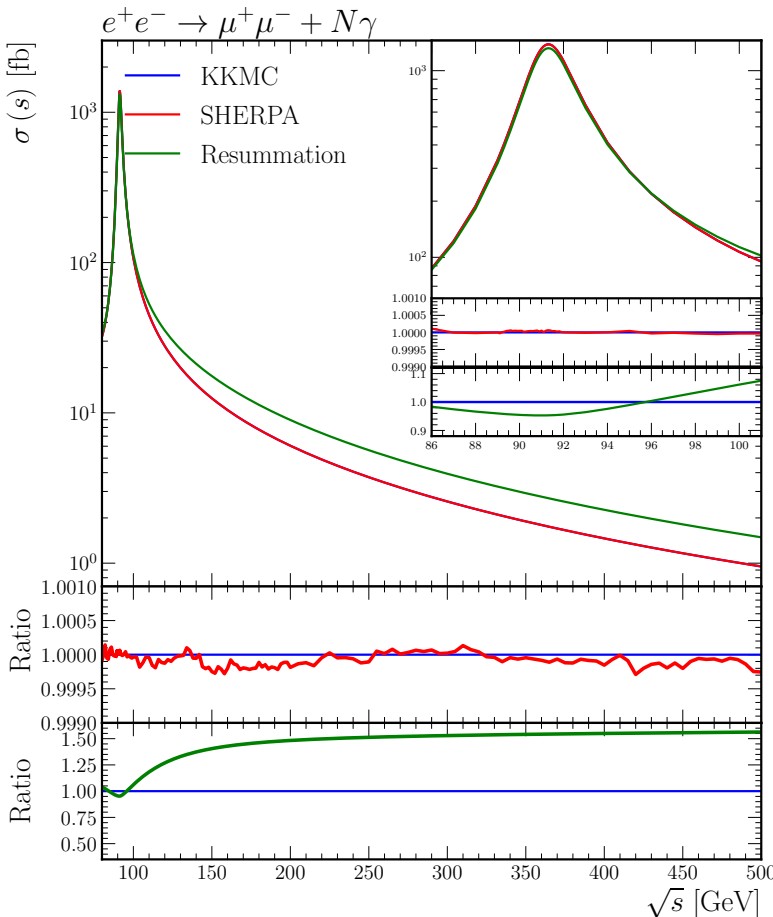

Figure 1: Total cross-section for $e^+e^- \to \mu^+\mu^-$ for 80GeV to 500GeV including ISR+FSR up to $\mathcal{O}(\alpha^3 L^3)$. In the first subplot we show the deviation from the KKMC generator at the same accuracy. In the second subplot, we compare the $\mathcal{O}(\alpha^3 L^3)$ against the pure resummed prediction. For the resummation only result all higher–-order $\tilde{\beta}$ have been set to zero and only $\tilde{\beta}_0^0 = \left|\mathcal{M}_0^0\right|^2$ has been kept.

# 3 Validation: $e^-e^+ \to f\bar{f}$ and $e^-e^+ \to W^-W^+$

In this section we will compare our results with other implementations of the YFS method. In particular, we will focus on the $e^+e^-$ generators KKMC [46,47] and KORALW/YFSWW [48]. As we are examining the effects of multiple soft photon emissions, their QED coupling should be evaluated in the Thompson limit, making $\alpha(0)$ the natural choice. We decouple this from scheme choices for electroweak parameters which we keep process-dependent, allowing in particular to use $\alpha$ at more suitable scales for the hard scatter.

$e^-e^+ \to f\bar{f}$

For this process we employ the $\alpha(0)$ electroweak scheme, with values for its input parameters listed in Table 3. In this scheme the widths of the gauge bosons are taken to be fixed, $\sin\theta_W$

Table 3: Electroweak input parameters in the $\alpha(0)$ scheme.

|  | Mass [ GeV ] | Width [ GeV ] |
|---|---|---|
| Z | 91.1876 | 2.4952 |
| W | 80.385 | 2.09692 |
| H | 125 | 0.00407 |
| e | 0.000511 | - |
| $\mu$ | 0.105 | - |
| $\alpha_{\mathrm{QED}}^{-1}(0)$ | 137.03599976 |  |

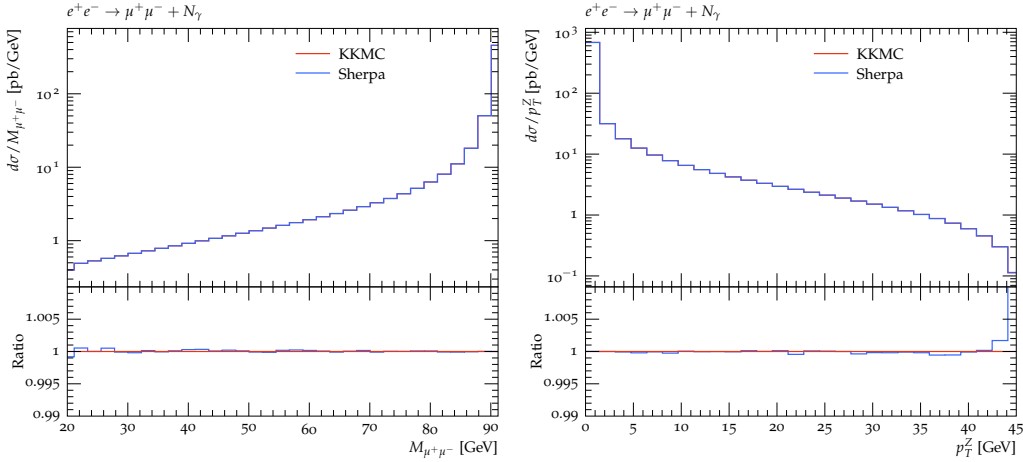

Figure 2: Plot of the invariant mass distribution of the final state particles. The nominal plots from SHERPA are displayed in the main frame. The sub-plots are the ratios with respect to KKMC.

is calculated from the W and Z masses as $\sin\theta_W = \sqrt{1 - M_W^2/M_Z^2}$. In general, all couplings associated with the emissions of soft photons are evaluated as $\alpha(0)$, while the couplings associated with hard process can be chosen to be different, for example $\alpha(M_Z)$ for fermion-pair production on the $Z$-pole. In the comparison between SHERPA and KKMC, we first compared the total $e^-e^+ \to \mu^-\mu^+$ cross sections from both codes at a wide range of energies, with results presented in Fig. 1. In these plots both ISR and FSR have been included as well as perturbative fixed–order corrections up to $\mathcal{O}\left(\alpha^3 L^3\right)$. We used a minimal phasespace cut of $M_{\mu^+\mu^-} > 20$GeV, and we took the infrared cut-off for photon energies to be $E_{cut}^\gamma = 0.1$MeV. Both electron and muon masses are included, as required by the YFS formalism. To facilitate a like–for–like comparison between our implementation and KKMC, we use the same input parameters and cuts, and turn off any initial-final interference (IFI) effects in KKMC; consequently, the KKMC matrix elements are taken in the exclusive exponentiation (EEX).

In Figs. 2 and 3 we plot some differential distributions that further elucidate the dynamics of soft photon emissions and verify that results obtained from our implementation agree with the corresponding KKMC results. In Fig. 2 we exhibit the invariant mass and $p_T$ distributions of the final state leptons, and we again find agreement at the permil level or below. At large values of the transverse momentum we notice small deviations between the two generators

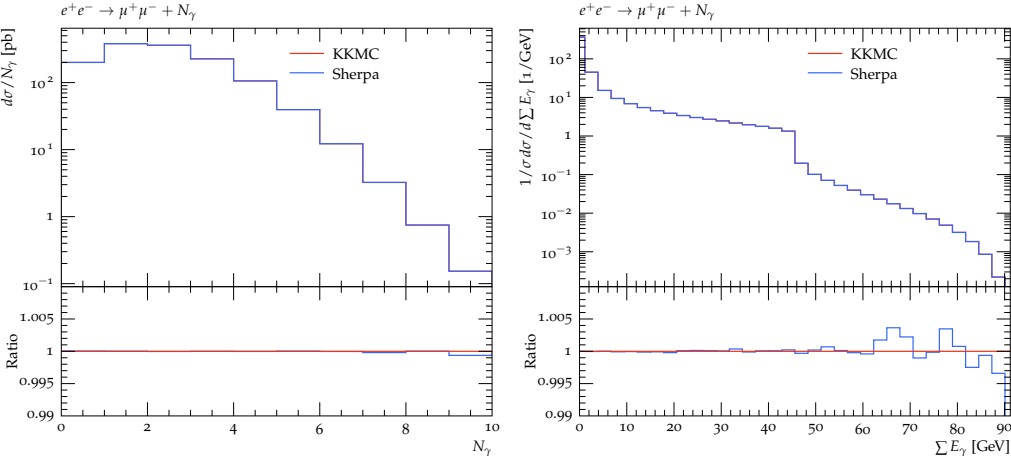

Figure 3: The total number of photons for both ISR and FSR (left), and the total sumeed photon energy (right), comparing SHERPA with KKMC.

attributable to a lack of statistics in this highly suppressed phase space region, which requires multiple non collinear photon emissions. In particular the $p_T$ plot shows the fully exclusive nature of the YFS scheme, in contrast to the more inclusive structure function approach, where the lepton pair transverse momentum is integrated out and could be recovered only *a posteriori*, for example, by combining the calculation with a parton shower.

The left plot in Fig. 3 compares the number of ISR+FSR photons, generated in SHERPA and in KKMC, and we again find excellent agreement at the sub-permil level. In the right part of the same figure, we show the summed photon energies $\sum_n E_\gamma$, which exhibits a distinct shape at $\frac{\sqrt{s}}{2}$. This shape results from the fact that the energy of a single photon is constrained to be less than $\frac{\sqrt{s}}{2}$, to exceed this kinematic limit the emission of at least two hard photons is needed. As before, we notice a slight statistical fluctuation in the tail of the photon-energy distribution.

## $e^-e^+ \to W^-W^+$

Dedicated tools for the calculation of $W^+W^-$ production cross sections fall into two categories: The first, including for example RACOONWW [58–60] treat ISR effects through the structure function approach, while in the second, exemplified by KORALW/YFSWW [48], ISR is treated using YFS resummation. Both of these tools include the full matrix elements for four–fermion production, $e^-e^+ \to 4f$, and KORALW/YFSWW also incorporates diagrams where the four fermions are produced through pairs of $Z$ bosons.

The infrared factorization of soft-photon emissions can be easily extended to include photon radiation off on-shell $W$ bosons. This is achieved by simply replacing the mass $m_f$ and charge $Q_f$ of spin-$\frac{1}{2}$ fermions in the amplitude and, correspondingly, the YFS form factor Eq. (A.1), with the mass $m_W$ and the charge $Q_W$ of the $W$ bosons [61]. In the case of $e^-e^+ \to W^-W^+$ these corrections induce an additional term in the YFS form factor, namely

$$Y(p_{e^+}, p_{e^-}) \to Y(p_{e^+}, p_{e^-}) + Y(p_{W^+}, p_{W^-}). \tag{16}$$

As we focus on the EEX version of the soft photon resummation, we neglect any coherence effects. One such effect occurs when a charged final state particle becomes close to a beam particle with the same, or very similar, charge such that the sum of initial and final state charges approaches 0. In these configurations the charges screen each other, and the pattern of photon emissions becomes increasingly similar to that expected from electric dipoles. As

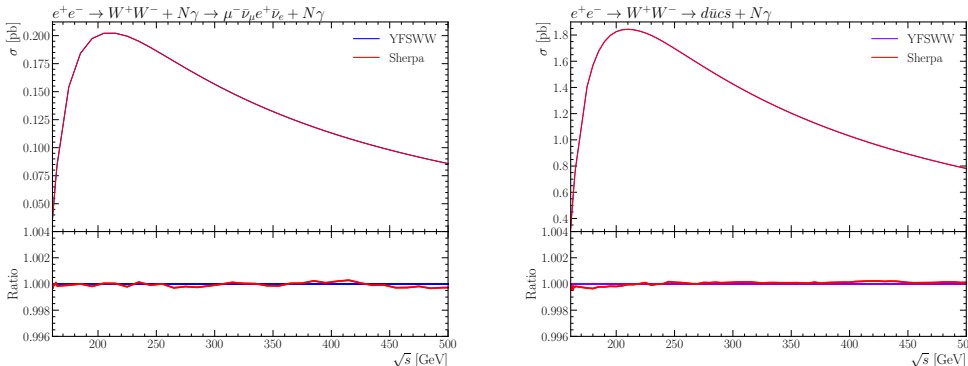

Figure 4: Total cross-section for hadronic and leptonic decay channel of $e^+e^- \to W^+W^-$.

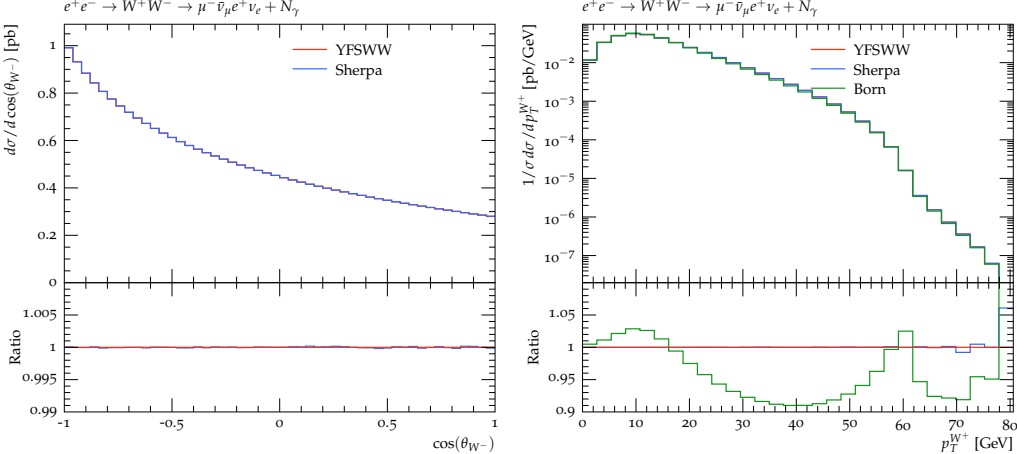

Figure 5: $\cos\Theta_{W^-}$ of the $W^-$ boson with respect to the incoming $e^-$ beam (left), and the corresponding transverse momentum (right), comparing SHERPA with YFSWW and the Born-level calculation.

a result the emitted photon spectrum becomes softer, an effect pointed out *e.g.* in [62]. To correctly capture such effects a fully coherent description of $e^+e^- \to 4f$ is necessary, which is beyond the scope of this paper. However, this effect can be safely neglected in cases where the particles are well separated, for example in the case of narrow resonant production.

Another important correction to $W^+W^-$ pair production cross sections stems from electromagnetic Coulomb interactions [63–65] at the production threshold, which leads to large corrections of the form $\approx \frac{\alpha\pi}{\beta_W}$, *i.e.* an enhancement by the inverse $W$-boson velocity $\beta_W$, which of course tends to zero at threshold. The analytic result for the all order Coulomb correction was presented in [66] and the effects of the second order contribution was investigated and shown to yield a positive correction to the production cross section in the threshold region. Being loop-induced this correction however is already included in the virtual part of the YFS form-factor once QED emissions from the $W$ bosons are included. Since the Coulomb effect can be treated to even higher orders we keep it as a separate contribution in SHERPA and therefore subtract it consistently from the YFS form factor. Following [61] this results in a subtracted

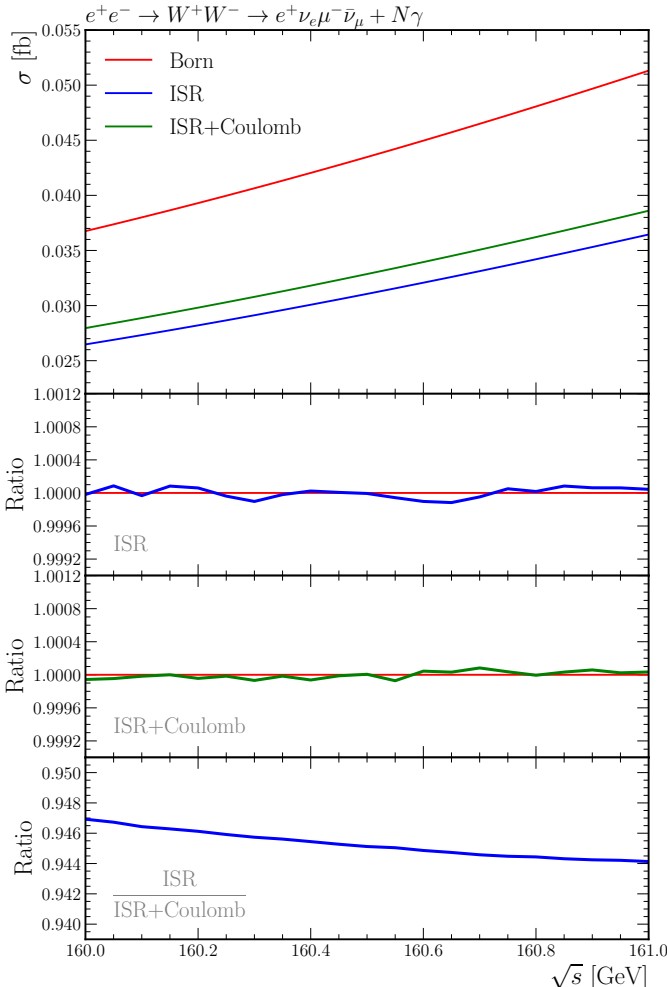

Figure 6: The effect of ISR and the Coulomb exchange at the $WW$ threshold, with the ratio taken with respect to YFSWW. The ISR reduces the total cross-section by about 30% and the Coulomb correction enhancement is of order 5%.

virtual contribution, namely

$$
\begin{aligned}
B_{\text{Subtracted}}(p_{W^+}, p_{W^-}) &= B_{\text{Full}}(p_{W^+}, p_{W^-}) - \frac{\theta(\beta_{\text{cut}} - \beta_W)}{\beta_W} \lim_{\beta_W \to 0} \beta B_{\text{Full}}(p_{W^+}, p_{W^-}) \\
&= B_{\text{Full}}(p_{W^+}, p_{W^-}) - \frac{\pi}{4\beta_W} \theta(\beta_{\text{cut}} - \beta_W). \quad (17)
\end{aligned}
$$

As made explicit in the equation above, the Coulomb correction can be extracted from the full virtual YFS form factor by taking a suitable limit, which is then subtracted in a region of small $\beta_W < \beta_{\text{cut}}$, the phase space regime where the Coulomb correction dominates. $\beta_{\text{cut}}$ can be taken as a user input and for the default value we follow the choice of KORALW/YFSWW with $\beta_{\text{cut}} = 0.382$.

For the validation of the SHERPA implementation we employed the $G\mu$ scheme including all particle masses, Table 4 for the specific values we used. For the comparison with KORALW we only consider the contribution of three diagrams, the $Z/\gamma$ s-channel and the neutrino mediated t-channel, to the four-fermion final state [67] [2]. In Fig. 4 we exhibit the total cross-sections for

---

[2]Of course the full set of diagrams can be easily included within the SHERPA framework, and first order real and

Table 4: List of electroweak parameters in the $G_\mu$ scheme used for the validation of SHERPA with KORALW in $e^-e^+ \to W^-W^+$ production processes.

|  | Mass [ GeV ] | Width [ GeV ] |
|---|---|---|
| Z | 91.1876 | 2.4952 |
| W | 80.385 | 2.09692 |
| H | 125 | 0.00407 |
| e | 0.000511 | - |
| $\mu$ | 0.105 | - |
| $\alpha_{G_\mu}^{-1}$ | 132.231948 |  |
| $G_F$ | $1.16637 \times 10^{-5} \mathrm{GeV}^{-2}$ |  |

hadronic and leptonic final states at different c.m. energies, obtained by multiplying the $WW$-pair production cross section with corresponding branching ratios. Again we see excellent agreement between SHERPA's YFS implementation and the KORALW benchmark. In terms of differential distributions, we see from Fig. 5 that both the $\cos \Theta_{W^-}$ and the transverse momentum distribution agree between both calculations. We also include the transverse momenta distribution associated with born calculation. This emphasises the effect of the ISR has on the $p_T$ which can be as large as 10%. It should be emphasised that for the ISR treatment in $e^+e^- \to W^+W^-$ nothing has changed in the resummation algorithm, demonstrating that the SHERPA implementation is independent of the final state. In Fig. 6 we demonstrate the impact of including the Coulomb correction around the threshold region. As expected, it provides a small, but non-negligible correction to the cross section in the threshold region which must be taken into account at future $e^+e^-$ colliders. It is also noteworthy that our current implementation does not take into account the effect of the sizeable $W$ width, or, conversely, its limited lifetime. This will be left to a future publication.

## 4 YFS vs Collinear Resummation

In this section we will present a comparison of the YFS treatment of QED initial-state radiation with its description in the collinear resummation or structure function approach [30], that has been traditionally used in MC tools such as SHERPA or WHIZARD. Instead of an expansion of the radiation pattern around the soft region in the YFS approach, the stucture function method focuses on large collinear logarithms and resums the corresponding leading-log (LL) terms using the DGLAP evolution equations [31–34]. Combining these evolution equations with leading-order (LO) initial conditions, trivially the Dirac-$\delta$ function, yields the structure functions. Going beyond LO, the trivial $\delta$ function is replaced with a more complex mixed electron-photon state, giving rise to the recently derived NLO structure functions [70–72]. These new PDFs are poised to become the new standard in MC tools, which will systematically include NLO corrections, for example in future versions of MadGraph5_aMC@NLO [73].

In any case, at the currently mainly used LL accuracy, the structure functions are convoluted with the "partonic" cross section as

$$d\sigma(s) = \int dx_1 dx_2 \, f_{e^+}(x_1, Q^2) f_{e^-}(x_2, Q^2) \, d\hat{\sigma}(x_1 x_2 s), \qquad (18)$$

where $d\hat{\sigma}$ is the differential partonic cross section. At this accuracy level, $f_{e^+}(x) = f_{e^-}(x)$, and

---

virtual corrections can be extracted from SHERPA's in-built matrix element generators or from automated NLO tools such as OPENLOOPS [68] or RECOLA [69].

Table 5: The different scheme choices available in SHERPA for the electron structure function, where $\beta = \frac{\alpha}{\pi}\left(\ln(s/m_e^2) - 1\right)$ and $\eta = \frac{\alpha}{\pi}\ln(s/m_e^2)$.

| Scheme | $\beta_S$ | $\beta_H$ | Refs |
|--------|-----------|-----------|------|
| Beta | $\beta$ | $\beta$ | [74] |
| Eta | $\eta$ | $\eta$ | [75] |
| Mixed | $\beta$ | $\eta$ | [76] |

it is given by

$$f_{e^{\pm}}(x, Q^2) = \beta \, \frac{\exp\left(-\gamma_E \beta + \frac{3}{4}\beta_S\right)}{\Gamma(1+\beta)}(1-x)^{\beta-1} + \beta_H \sum_{n=0}^{\infty} \beta_H^n \, \mathcal{H}_n(x), \tag{19}$$

with $\beta = \frac{\alpha}{\pi}\left(\ln(s/m_e^2) - 1\right)$. This form of $\beta$ is a direct result of the analytical integration over the entire soft-photon phasespace and any change to its definition will cause the IR subtraction to fail. There is however some residual freedom in how non-leading terms are taken into account, reflected in the treatment of the soft photon residue $\beta_S$ as either $\ln(s/m_e^2)$ or $\ln(s/m_e^2)-1$. This contrasts with the treatment in YFS resummation, where this freedom is absent. The hard coefficients $\mathcal{H}_n(x)$ are given in [74–77]. By default, SHERPA includes the hard coefficients up to second order ($n = 2$). For some popular choices of the various $\beta$'s, *cf.* Table 5. When undertaking a comparison between the YFS resummation and the collinear resummation care should be taken about which terms are included. For example, if we consider the form factor in Eq. (19) we see that it is LL in nature while if we consider the YFS form factor it contains terms that are at NLL. This can be easily seen if we take the high energy limit of the form factor,

$$Y(\Omega_I) \underset{s \gg m_e^2}{\Longrightarrow} \exp\left\{\beta\left[\log(\frac{2k_{\min}}{\sqrt{s}}) + \frac{1}{4}\right] + \frac{\alpha}{\pi}\left(\frac{\pi^2}{3} - \frac{1}{2}\right)\right\}. \tag{20}$$

The numerical effect of this addition can be seen in Fig. 7 were we consider the YFS cross-section with and without this NLL contribution. We see that the agreement between the strictly LL YFS calculation and the collinear resummation is around 0.1% when the full YFS calculation is used, including the NLL terms of soft origin, the difference is around 0.6%.

## 5 Higgs Production

We finally turn to a first application of our implementation for a new case, namely the production of Higgs bosons in the Higgs-Strahlungs process at a future electron–positron collider, $e^-e^+ \to ZH$. To frame the discussion, we first depict in Fig. 8 the c.m.-energy dependent Higgs production cross section in various channels, and compare the results at Born level with those obtained by including QED ISR in the YFS scheme. We see, as expected, that QED ISR tends to increase the cross sections at high c.m.-energies in those processes that are driven by the exchange of an $s$-channel propagator, such as $e^+e^- \to ZH$ and the top-associated production, while it tends to decrease the cross section at their peaks. The increase can be thought off as the effect of some form of a "radiative return" to the peak, while the decrease at the peak can be understood as a "washing out" of the large peak cross section by reducing the c.m.-energy due to the ISR. Conversely, QED ISR decreases the production cross sections throughout in those processes that are $t$-channel dominated.

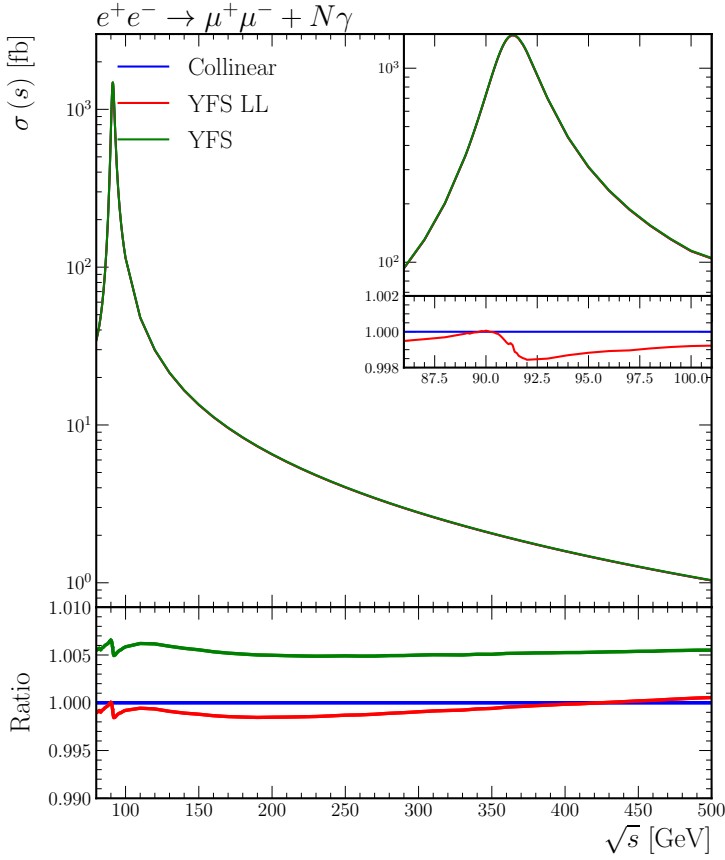

Figure 7: Total cross-section for $e^+e^- \to \mu^+\mu^-$ for 80GeV to 500GeV where the ISR is modelled using a collinear (blue) resummation and compared against the soft resummation up to $\mathcal{O}\left(\alpha^3 L^3\right)$. The red line represents the strictly leading logarithmic YFS resummation while the green represents the full YFS calculation.

In Fig. 9 we validate, once more, the treatment of QED FSR in our implementation, by comparing the invariant mass distribution of the muon pair in $e^-e^+ \to Z_{\to\mu^-\mu^+}H$ obtained at Born level and by adding FSR in two independent implementations of the YFS scheme within SHERPA. The muons are defined at the dressed level, meaning that nearby photons are recombined with the undressed momenta of the muons using a cone size of $\Delta R = 0.1$. We apply a phasespace cut on the final state muons satisfying $80 \leq M_{\mu^-\mu^+} < 115$. The electroweak parameters are the same that we used in the KKMC validation and are summarized in Table 3. At the perturbative level we note that there are slightly different approaches of how the $\tilde{\beta}$ terms are implemented. In PHOTONS these corrections are calculated using matrix element corrections to the bosonic decays. These are exact corrections and are constructed using spin amplitude methods. The corrections we employ for the PHOTONS calculation are NLO accurate in QED, which corresponds to the second row of Table 2. As a validation of our new FSR implementation these are also the $\tilde{\beta}$ terms we keep for the YFS++ calculation. We see the by now familiar form of the shift of the mass distribution once FSR is added, and in addition we confirm that our two implementations in the established PHOTONS module [44] and in our new YFS++ module are in perfect agreement. It is worth noting that we normalised the distributions

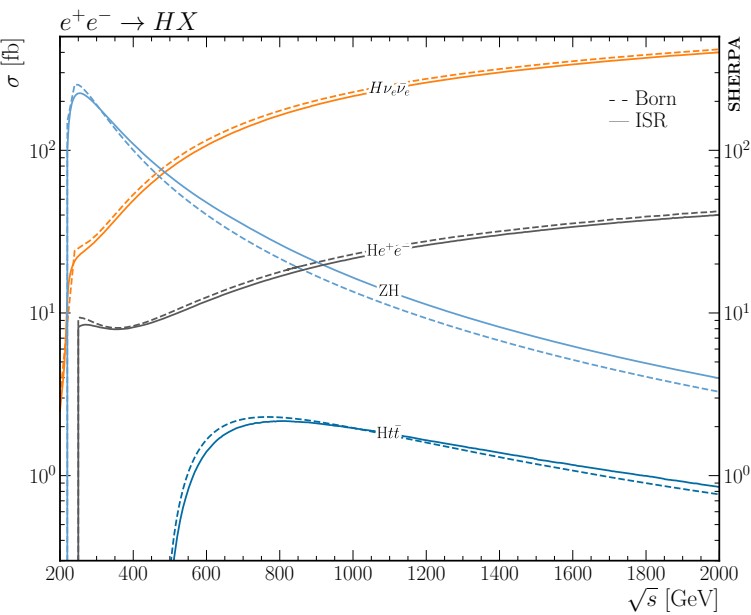

Figure 8: Total cross-section for $e^+e^- \to HX$ at the Born level and with ISR corrections included. For the ISR we include the perturbative corrections up to $\mathcal{O}\left(\alpha^3 L^3\right)$

on the production cross section, which is necessary because in YFS++ both QED ISR and FSR are included, while PHOTONS is specifically geared towards inclusion of QED radiation in decays only.

# 6 Summary and Outlook

In this publication we have described, in detail, the implementation of soft photon resummation in the Yennie–Frautschi–Suura scheme for initial and final state emissions in a new module YFS++ within the SHERPA event generator. To take maximal advantage of the calculation tools already existing in SHERPA we have restricted our implementation to the EEX approach, and we anticipate that we will fully combine our new framework with automated modern spin-amplitude methods, for example Berends-Giele recursion [78], to calculate the IR finite $\tilde{\beta}$ using SHERPA's built-in ME generators, AMEGIC [79] and COMIX [80].

We validated our new implementation by detailed comparison with established tools. In particular we confirmed the correctness of our implementation in $e^-e^+ \to \mu^-\mu^+$ by comparison with KKMC, where we found perfect agreement between the two codes. We also checked our code in a second process, $e^-e^+ \to W^-W^+$. While we focused mainly on the effect of QED initial state radiation, to confirm the process–independence of our implementation, we also added the Coulomb correction to the $W$-pair production at threshold. We confirmed the correctness of our results by finding perfect agreement with corresponding results obtained from KORALW. In a second validation step we confirmed, again for $e^-e^+ \to \mu^-\mu^+$, that the treatment of QED ISR in the YFS approach yields results that are comparable to those obtained by using the structure function approach. This is an important finding, as it shows that expansion of the radiation pattern around either soft or collinear regions of emission phase space and the subsequent resummation of the emerging corresponding large logarithms yield numerically consistent results.

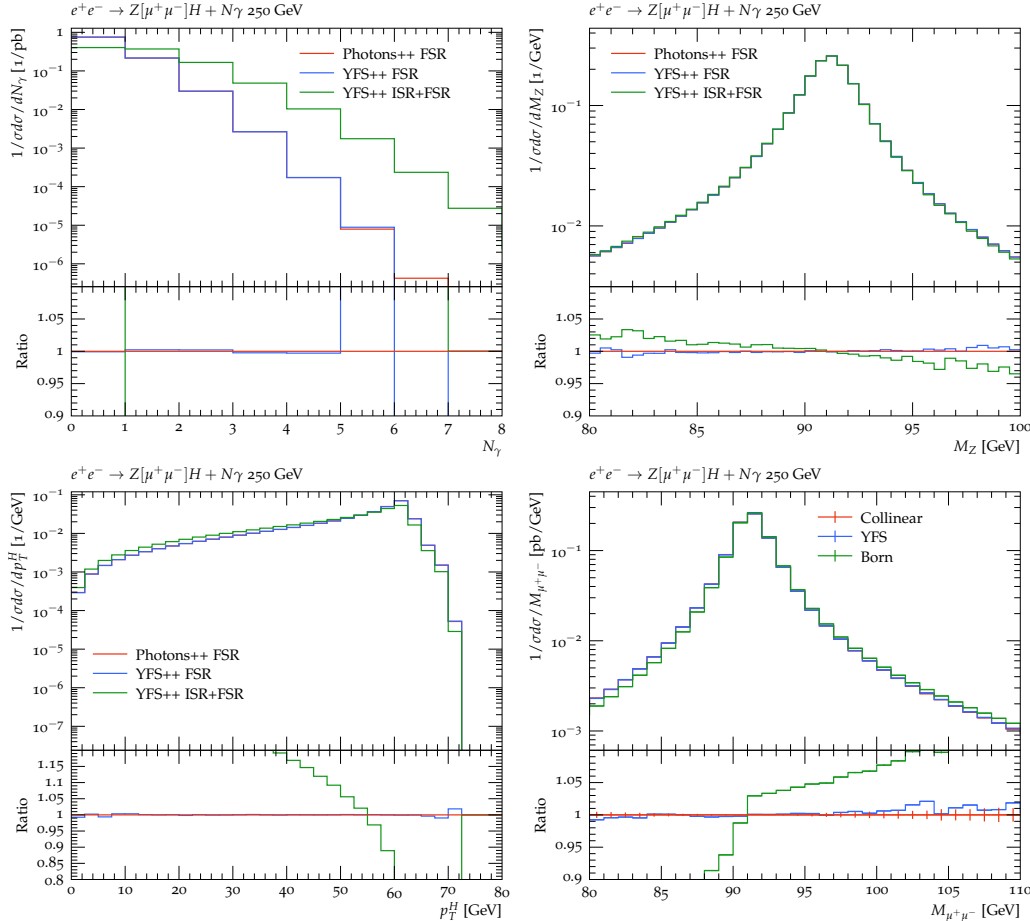

Figure 9: Photon multiplicity for PHOTONS module and the new implementation (top left), contrasting pure FSR (red and blue) as well the combined ISR+FSR (Green), the reconstructed $Z$ resonance mass (top right), and the Higgs transverse momentum (bottom left). The reconstructed $Z$ mass, again, contrasting the impact of ISR as modelled in collinear factorisation and YFS resummation (bottom right) as further validation.

We finally turned to produce a first new result, and analysed the impact of QED radiation in the YFS scheme on Higgs production in the Higgs-Strahlung process at electron–positron colliders. This further shows that our implementation can be applied in a process independent way, rendering it an interesting asset for future precision studies of important processes at the planned lepton colliders of the next decades.

We intend to build on this successful implementation and extend it in various ways. As already indicated, as a first step we plan to fully connect the YFS resummation with automated fixed–order tools to automatically include at least the first–order/one–loop corrections. We hope that we can further improve the accuracy for important processes by adding even higher–order corrections where possible. In addition, we realise that the treatment of unstable particles, *i.e.* resonances with sizable widths, deserves special consideration. We plan to build on the important work by Jadach *et al.* in [56] and extend their treatment to other processes. It is anticipated that the features described in this paper will be made available in the future SHERPA 3 release.

# Acknowledgements

A.P would like to thank Stanislaw Jadach for his help throughout all stages of this work. This work has received funding from the European Union's Horizon 2020 research and innovation programme as part of the Marie Skłodowska-Curie Innovative Training Network MCnetITN3 (grant agreement no. 722104) and the Marie Skłodowska-Curie grant agreement No 945422. F.K. gratefully acknowledges funding as Royal Society Wolfson Research fellow. M.S. is funded by the Royal Society through a University Research Fellowship (URF\R1\180549) and an Enhancement Award (RGF\EA\181033 and CEC19\100349).

# A   YFS Infrared Functions

In this appendix, we present some analytical representations of the YFS infrared (IR) functions corresponding to the emission of virtual and real photons from a pair of charged massive particles. The YFS-Form-Factor $Y(\Omega)$ reads

$$Y(\Omega) = 2\alpha \sum_{i<j} \left( \mathcal{R}e\, B(p_i, p_j) + \tilde{B}(p_i, p_j, \Omega) \right) ,$$

where the virtual eikonal factor is given by

$$B(p_i, p_j) = -\frac{i}{8\pi^3} Z_i Z_j \theta_i \theta_j \int \frac{\mathrm{d}^4 k}{k^2} \left( \frac{2p_i \theta_i - k}{k^2 - 2(k \cdot p_i)\theta_i} + \frac{2p_j \theta_j + k}{k^2 + 2(k \cdot p_j)\theta_j} \right)^2 ,$$

and the real eikonal factor reads

$$\tilde{B}(p_i, p_j, \Omega) = \frac{1}{4\pi^2} Z_i Z_j \theta_i \theta_j \int \mathrm{d}^4 k\, \delta(k^2)(1 - \Theta(k, \Omega)) \left( \frac{p_i}{(p_i \cdot k)} - \frac{p_j}{(p_j \cdot k)} \right)^2 .$$

Here $Z_i$ and $Z_j$ are the charges of particles $i$ and $j$ in units of the positron charge, respectively, and $\theta_{i,j} = \pm 1$ for final (initial) state particles. $\Omega$ is the region of the phase space for which the soft photons cannot be resolved. The divergences present in this expression need to be regularised, which can be achieved by either introducing a fictitious small photon mass $m_\gamma$, as in the original YFS paper [43], or through dimensional regularisation.

**Virtual IR Function**

Here we present the expression for the virtual part of the YFS for any two charged massive particles.

$2\alpha \mathcal{R}B(p_1, p_2)$

$$= -Z_i Z_j \theta_i \theta_j \frac{\alpha}{\pi} \Bigg[ \left( \frac{1}{\rho} \ln \frac{\mu(1+\rho)}{m_1 m_2} - 1 \right) \ln \frac{m_\gamma^2}{m_1 m_2} + \frac{\mu\rho}{s} \ln \frac{\mu(1+\rho)}{m_1 m_2} + \frac{m_1^2 - m_2^2}{2s} \ln \frac{m_1}{m_2} - 1$$

$$+ \frac{1}{\rho} \left( \pi^2 - \frac{1}{2} \ln \frac{\mu(1+\rho)}{m_1^2} \ln \frac{\mu(1+\rho)}{m_2^2} - \frac{1}{2} \ln^2 \frac{m_1^2 + \mu(1+\rho)}{m_2^2 + \mu(1+\rho)} - \mathrm{Li}_2(\zeta_1) - \mathrm{Li}_2(\zeta_2) \right) \Bigg],$$

$$(A.1)$$

where,

$$\mu = p_1 p_2, \quad s = 2\mu + m_1^2 + m_2^2, \quad \rho = \sqrt{1 - \left( \frac{m_1 m_2}{\mu} \right)^2}, \quad \zeta_i = \frac{2\mu\rho}{m_i^2 + \mu(1+\rho)}. \qquad (A.2)$$

**Real IR Function**

Here we present an expression for the IR function $\tilde{B}$ which corresponds to the emission of a real photon $k \in \Omega$ from a dipole consisting of two charged massive particles $p_1$ and $p_2$.

$$
2\alpha\tilde{B}(p_1, p_2) = -Z_i Z_j \theta_i \theta_j \frac{\alpha}{\pi} \bigg[ \left( \frac{1}{\rho} \ln \frac{\mu(1+\rho)}{m_1 m_2} - 1 \right) \ln \frac{\omega}{m_\gamma^2} + \frac{1}{2\beta_1} \ln \frac{1+\beta_1}{1-\beta_1}
$$
$$
+ \frac{1}{2\beta_2} \ln \frac{1+\beta_2}{1-\beta_2} + \mu G(p_1, p_2) \bigg], \tag{A.3}
$$

where $\beta_i = \frac{|\vec{p}_i|}{E_i}$ and $\mu, \rho$ are defined in Eq. (A.2) and $\omega$ is the momentum cut-off specifying $\Omega$ in the frame $\tilde{B}$ is to be evaluated in. $G(p_1, p_2)$ is a complicated function that can be expressed as a combination of logarithms and dilogarithms,

$$
G(p_1, p_2) = \frac{1}{\sqrt{(Q^2+M^2)(Q^2+\delta^2)}} \bigg[ \ln \frac{\sqrt{\Delta^2+Q^2}-\Delta}{\sqrt{\Delta^2+Q^2}+\Delta} \left[ \chi_{23}^{14}(\eta_1) - \chi_{23}^{14}(\eta_0) \right] + Y(\eta_1) - Y(\eta_0) \bigg], \tag{A.4}
$$

where,

$$
\chi_{kl}^{ij} = \ln \left| \frac{(\eta-y_i)(\eta-y_j)}{(\eta-y_k)(\eta-y_l)} \right|,
$$
$$
Y(\eta) = Z_{14}(\eta) + Z_{21}(\eta) + Z_{32}(\eta) - Z_{34}(\eta) + \frac{1}{2} \chi_{34}^{12}(\eta) \chi_{14}^{23}(\eta),
$$
$$
Z_{ij}(\eta) = 2\text{Li}_2\left( \frac{y_j-y_i}{\eta-y_i} \right) + \frac{1}{2} \ln^2 \left| \frac{\eta-y_i}{\eta-y_j} \right|, \tag{A.5}
$$

and

$$
\eta_0 = \sqrt{E_2^2-m_2^2}, \eta_1 = \sqrt{E_1^2-m_1^2} + \sqrt{\Delta^2+Q^2},
$$
$$
y_{1,2} = \frac{1}{2}\bigg[ \sqrt{\Delta^2+Q^2} - E + \frac{M\delta \pm \sqrt{(Q^2+M^2)(Q^2+\delta^2)}}{\sqrt{\Delta^2+Q^2}+\Delta} \bigg],
$$
$$
y_{3,4} = \frac{1}{2}\bigg[ \sqrt{\Delta^2+Q^2} + E + \frac{M\delta \pm \sqrt{(Q^2+M^2)(Q^2+\delta^2)}}{\sqrt{\Delta^2+Q^2}-\Delta} \bigg], \tag{A.6}
$$

where the following notation has been introduced,

$$
\Delta = E_1 - E_2, E = E_1 + E_2,
$$
$$
\delta = m_1 - m_2, M = m_1 + m_2,
$$
$$
Q^2 = -(p_1-p_2)^2. \tag{A.7}
$$

## B  Photon Generation

In this section we discuss the explicit construction of the photon momenta in the Monte Carlo integration and event generation. In each event the overall number of resolved photons is generated according to a Poissonian with mean

$$
\langle n_\gamma \rangle = -\frac{\alpha}{\pi} \ln\left( \frac{E_{\text{max}}}{E_{\text{min}}} \right) \left( \frac{1+\beta_1\beta_2}{\beta_1+\beta_2} \ln \frac{(1+\beta_1)(1+\beta_2)}{(1-\beta_1)(1-\beta_2)} - 2 \right), \tag{B.1}
$$

where $\beta_i = \frac{|\vec{p}_i|}{p_i^0}$. We introduce an explicit form for the photon momentum parameterizing the momentum using polar coordinates.

$$k_i = \frac{\sqrt{s}}{2} x_i (1, \sin\theta_i \cos\phi_i, \sin\theta_i \sin\phi_i, \cos\theta_i). \tag{B.2}$$

In this parameterization the eikonal term is given by,

$$\frac{d^3 k_i}{k_i^0} \tilde{S}(k_i) = \frac{dx_i}{x_i} d(\cos\theta_i) d\phi_i \, g(\theta_i), \tag{B.3}$$

where,

$$\tilde{S}(k) = \sum_{i,j} \frac{\alpha}{4\pi^2} Z_i Z_j \theta_i \theta_j \left( \frac{p_i}{p_i \cdot k} - \frac{p_j}{p_j \cdot k} \right)^2 \tag{B.4}$$

and $g(\theta)$ is given by

$$g(\theta) = \frac{\alpha}{\pi^2} \left( \frac{2(1+\beta_1\beta_2)}{(1-\beta_1\cos\theta)(1+\beta_2\cos\theta)} - \frac{1-\beta_1^2}{(1-\beta_1\cos\theta)^2} - \frac{1-\beta_2^2}{(1+\beta_2\cos\theta)^2} \right). \tag{B.5}$$

**Photon Angles**

There are two angles used in the parametrisation of the photon momentum that have to be generated. The first, $\phi$, is trivial and is given by,

$$\phi = 2\pi\#, \tag{B.6}$$

where $\#$ is a uniformly generated random number $\in (0,1)$. The remaining angle $\theta$ is slightly more complex. It is generated by sampling from the $\tilde{S}(k)$ distribution,

$$\begin{aligned}
\tilde{S}(k) &\propto \left( \frac{p_1}{p_1 k} - \frac{p_2}{p_2 k} \right)^2 \\
&= \left( \frac{p_1^2}{(p_1 k)^2} + \frac{p_2^2}{(p_2 k)^2} - \frac{2p_1 p_2}{(p_1 k)(p_2 k)} \right).
\end{aligned} \tag{B.7}$$

By taking $p_1$ and $p_2$ to be the beam momenta, *i.e.* $p_{1/2} = \left( \frac{\sqrt{s}}{2}, 0, 0, \pm p_z \right)$, the eikonal is given by

$$\tilde{S}(k) \propto \left( \frac{1-\beta_1^2}{(1-\beta_1\cos\theta)^2} + \frac{1-\beta_2^2}{(1-\beta_2\cos\theta)^2} - \frac{2(1+\beta_1\beta_2)}{(1-\beta_1\cos\theta)(1+\beta_2\cos\theta)} \right). \tag{B.8}$$

The interference term can be rewritten as,

$$\frac{1}{(1-\beta_1\cos(\theta_i))(1+\beta_2\cos(\theta_i))} = \frac{\beta_1\beta_2}{\beta_1+\beta_2} \left( \frac{1}{\beta_2(1-\beta_1\cos(\theta_i))} + \frac{1}{\beta_1(1+\beta_2\cos(\theta_i))} \right). \tag{B.9}$$

Then $\cos(\theta_i)$ is generated according to either of the two terms in the interference. Generating it according to $\left( 1 \mp \beta_i \cos(\theta_i) \right)^{-1}$

$$\begin{aligned}
\int_{-1}^{y} d\cos(\theta_i) \frac{1}{1 \mp \beta_i \cos(\theta_i)} &= \# \int_{-1}^{1} d\cos(\theta_i) \frac{1}{1 \mp \beta_i \cos(\theta_i)}, \\
\cos(\theta_i) &= \frac{1}{\beta_i} \left( 1 - (1 \pm \beta_i) \left( \frac{1 \mp \beta_i}{1 \pm \beta_i} \right)^{\#} \right).
\end{aligned} \tag{B.10}$$

The probability associated with either of these distributions is given by,

$$P_i = \frac{\ln\left(\frac{1\pm\beta_i}{1\mp\beta_i}\right)}{\ln\left(\frac{1+\beta_1}{1-\beta_1}\right) + \ln\left(\frac{1+\beta_2}{1-\beta_2}\right)}.$$ (B.11)

The overall correction weight for this distribution is given by,

$$W = \frac{\dfrac{2(1+\beta_1\beta_2)}{(1-\beta_1\cos\theta)(1+\beta_2\cos\theta)} - \dfrac{1-\beta_1^2}{(1-\beta_1\cos\theta)^2} - \dfrac{1-\beta_2^2}{(1-\beta_2\cos\theta)^2}}{\dfrac{2(1+\beta_1\beta_2)}{(1-\beta_1\cos(\theta_i))(1+\beta_2\cos(\theta_i))}}.$$ (B.12)

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
