# Peer review of "YFS Resummation for Future Lepton-Lepton Colliders in SHERPA"

_SciPost Physics, doi:SciPost Phys. 13, 026 (2022)_

## Round 2 · Referee Report · Anonymous (Referee 1) · 2022-4-8

Report

The authors present a process-independent implementation of ISR and FSR soft-photon resummation in the Yennie-Frautschi-Suura (YFS) scheme within the SHERPA event-generation framework. Their development is intended to replace an existing collinear resummation framework in the structure-function approach in SHERPA, as a means to extend the accuracy of the resummation. As such, they present their work as a first step towards adapting the SHERPA event generator to the needs of future lepton-lepton colliders, where high-precision simulations of radiative QED corrections are of high importance. Supplemented by two appendices reviewing technical details, the article presents a thorough and detailed description of all aspects that are required to implement the YFS scheme in Monte-Carlo event generators in practice.

Anchored in the soft limit, YFS resummation neglects logarithmic enhancements arising from collinear singularities. After reviewing the theoretical basis of the YFS method, the authors argue that, up to a certain order, these can be included in the resummation via infrared-finite matrix-element corrections. They continue to highlight two different approaches to achieve this, dubbed EEX and CEEX. While the latter is stated to be theoretically superior, the EEX approach is pursued in their work. This is justified by a statement that the treatment of the CEEX approach is more difficult in practice, as it is formulated on the amplitude level. While the authors are of course free to choose the method that they deem more appropriate for their implementation, I would like to question the generality of this assessment. Naively, it does not seem a "veritable obstacle" to implement an amplitude-based approach, as the infrared singularity structure of photon radiation is known and understood on the amplitude level. If it includes interference effects that would otherwise be neglected, this might be a worthwhile investment, especially when an extension of the implementation to include initial-final radiation is foreseen. Is this statement meant to indicate that an amplitude-based approach is more intricate within SHERPA or are there other, more subtle, difficulties that arise in this construction? In either case, I would like to recommend a clarification of the rationale behind choosing the EEX scheme over the CEEX scheme.

Sections 3 and 4 provide a thorough validation of the YFS implementation through a comparison to existing tools for two processes on the one hand and a comparison to the existing collinear resummation in SHERPA for a single process on the other hand. The level of agreement with existing tools is an excellent verification of the correctness of their implementation and the cross check against collinear resummation reassures the viability of the YFS scheme compared to the structure-function approach. However, it seems like figure 5 lacks a discussion in the text, which I would ask the authors to add.

In my opinion, the core results are presented in section 5, where the new resummation framework is applied to Higgs production via Higgsstrahlung at a future electron-positron collider. The relevance of QED resummation for precision calculations in associated Higgs production is highlighted by presenting the total cross section for the most relevant production processes as a function of the centre-of-mass energy with and without ISR corrections. As a further validity check, the FSR implementation in the new YFS++ module is compared to the existing FSR implementation in the Photons++ module, again showing excellent agreement between the two. Referring to figure 8, the authors state that results are shown "in both the YFS scheme and the structure function approach". However, this does not seem to be the case, as figure 8 only contains two results for each process, one labelled "Born" and one labelled "ISR". I conclude that only the YFS result is presented. If this is true, the sentence in the text should be corrected to reflect the content of figure 8.

Finally, I have two general remarks that I suggest the authors consider addressing.

First, I could not find a statement on whether the new YFS module is already or may become part of a public SHERPA release in the future. A comment on this would be appreciated in light of the relevance for studies on future lepton-lepton facilities.

Secondly, I am wondering whether the authors have considered a comparison with QED parton showers, at least on the conceptual level. In particular, given that their resummation implementation treats additional photons similarly by explicitly adding them to the event record and that their resummation neglects initial-final interference effects, which are simulated in many QED showers. While it is certainly beyond the scope of this article to contrast the two approaches in detail (be it analytically or numerically), I was surprised that this point was entirely absent from their discussion.

In summary, the authors present an interesting piece of work and convincingly demonstrate the relevance for physics programs at future electron-positron colliders, which in my opinion meets the standards of SciPost Physics. I am therefore happy to recommend the article for publication once my remarks have been addressed.

The otherwise clear and well-written article contains a number of misprints and typos which should be corrected before publication: - page 2, towards the bottom of the page: left-over sentence "This run will produce 1 million ZH events with an ." - page 3, second paragraph: "International Linear (ILC)" -> "International Linear Collider (ILC)" - page 3, second paragraph: "Being smaller that CLIC" -> "Being smaller than CLIC" - page 3, second paragraph: probably missing macro in "[...] production of W pairs at threshold, eeww" - page 3, second paragraph: "Final runs at the $t\bar{t}$ and [...]" -> "Final runs at the $t\bar{t} threshold and [...]$" - page 3, third paragraph, below Table 1: words missing in "[...] to reduce the QED uncertaintites to an acceptable, Table 1" - page 4, last paragraph: words missing in "The indices $i$ and $j$ matrix elements [...]" - page 10, first paragraph: acronym IFI undefined - page 10, second paragraph: "per-mille" vs "permil" - page 11, Figure 4: left-over note to be deleted: "NOTE: More hadronic stats on the way" - page 12, Figure 5: misaligned minus sign in "$\cos\Theta_{W^-}$" - page 12, Figure 5: "[...] comparing SHERPA woth [...]" -> "[...] comparing SHERPA with [...]" - page 12, last paragraph: "CC03" undefined - page 13, Figure 6: repeated word in "The effect of of [...]" - page 14, first paragraph: "In Section 3 we exhibit [...]" should probably read "In Figure 4 we exhibit [...]" - page 14, first paragraph: "In Section 3 we demonstrate [...]" should probably read "In Figure 6 we demonstrate [...]" - page 14, second paragraph: "MadGraph_aMC@NLO" should be referred to as "MadGraph5_aMC@NLO" - page 14, third paragraph, right above Table 5: "For some popular choices of the various $\beta$'s, cf. Section 4" should probably read "For some popular choices of the various $\beta$'s, cf. Table 5" - page 16, first paragraph: "[...] collinear care should be taken [...]" should probably read "[...] collinear resummation care should be taken [...]" - page 16, first paragraph: missing word in "[..] in Equation (4.2) we see that it LL [...]" - page 16, second paragraph: "[...] we first depict in Fig. Figure 8 [...]" -> "[...] we first depict in Figure 8 [...]" - page 16, third paragraph: "In Fig. Section 5 we validate [...]" -> "In Figure 9 we validate [...]" - page 17, towards the bottom of the page: probably missing macro for YFS++ in "[...] these are also the $\tilde{\beta}$ terms we keep for the YFS++ calculation" - page 18, second paragraph: additional hyphen in "[...] to confirm the process-independence [...]" - page 19, third paragraph: repeated word in "[...] which corresponds to the the [...]" - page 19, third paragraph: slightly obscure reference to "Appendix A" (while being in Appendix A) - page 20, third paragraph: missing word in "[...] where # is uniformly generated random number [...]" - page 20, third paragraph: missing space between "$\tilde{S}(k)$" and "distribution"

  • validity: top
  • significance: high
  • originality: good
  • clarity: high
  • formatting: excellent
  • grammar: excellent

Author:  Alan Price  on 2022-06-20  [id 2596]

(in reply to Report 1 on 2022-04-08)
Category:
answer to question
correction

We would like to thank the referee for the careful and detailed consideration of our manuscript. Your comments have helped us improve the overall quality of the paper. The detailed replies and the corresponding changes to the draft are listed below. For the referees convenience a pdf file highlighting any changes made has been attached below.

[Referee] "While the authors are of course free to choose the method that they deem more appropriate for their implementation, I would like to question the generality of this assessment. Naively, it does not seem a "veritable obstacle" to implement an amplitude-based approach, as the infrared singularity structure of photon radiation is known and understood on the amplitude level. If it includes interference effects that would otherwise be neglected, this might be a worthwhile
investment, especially when an extension of the implementation to include initial-final radiation is foreseen. Is this statement meant to indicate that an amplitude-based approach is more intricate within SHERPA or are there other, more subtle, difficulties that arise in this construction? In either case, I would like to recommend a clarification of the rationale behind choosing the EEX scheme over the CEEX scheme."

[Reply] The choice of EEX scheme is reflective of the fact that it is technically easier to implement in an automatic fashion. While the singularity structure of QED is well known, there still exists some technical details that need to be resolved before we can make our implementation public. For example, the mandatory inclusion of lepton masses makes calculating one-loop diagrams, at the amplitude level, quite difficult.

[Referee] "Sections 3 and 4 provide a thorough validation of the YFS implementation through a comparison to existing tools for two processes on the one hand and a comparison to the existing collinear resummation in SHERPA for a single process on the other hand. The level of agreement with existing tools is an excellent verification of the correctness of their implementation and the cross check against collinear resummation reassures the viability of the YFS scheme compared to the structure-function approach.
However, it seems like figure 5 lacks a discussion in the text, which I would ask the authors to add."

[Reply] A discussion of figure 5 has been added to the text.

[Referee] "Referring to figure 8, the authors state that results are shown "in both the YFS scheme and the structure function approach". However, this does not seem to be the case, as figure 8 only contains two results for each process, one labelled "Born" and one labelled "ISR". I conclude that only the YFS result is presented. If this is true, the sentence in the text should be corrected to reflect the content of figure 8."

[Reply] It is correct that only the YFS results are present. The text has been corrected to reflect this.

[Referee] "I could not find a statement on whether the new YFS module is already or may become part of a public SHERPA release in the future. A comment on this would be appreciated in light of the relevance for studies on future lepton-lepton facilities."

[Reply] This module will be released in the upcoming Sherpa 3.X series. A sentence on this has been added to the conclusion.

[Referee] "I am wondering whether the authors have considered a comparison with QED parton showers, at least on the conceptual level. In particular, given that their resummation implementation treats additional photons similarly by explicitly adding them to the event record and that their resummation neglects initial-final interference effects, which are simulated in many QED showers.
While it is certainly beyond the scope of this article to contrast the two approaches in detail (be it analytically or numerically), I was surprised that this point was entirely absent from their discussion."

[Reply] The comparison of the YFS resummation versus a parton shower is indeed beyond the scope of this paper. In particular, the authors wish to do such a detailed study at the highest possible accuracy. For the parton shower this will require at least MC@NLO style study while for YFS we wish to, if possible, include an implementation of CEEX that is supplemented with full one-loop electroweak corrections.

[Referee] "The otherwise clear and well-written article contains a number of misprints and typos which should be corrected before publication"

[Reply] All misprints and typos have been corrected in the text.

Attachment:

diff-1-v3.pdf

---

## Round 3 · Referee Report · Anonymous (Referee 1) · 2022-6-28

Report
I would like to thank the authors for their response and for addressing my points of criticism.
While I am still surprised by the fact that they choose to omit any reference to QED showers in their discussion, I believe that this should not be a reason against publishing the article.
I would therefore like to recommend this paper for publication in SciPost Physics.
While I am still surprised by the fact that they choose to omit any reference to QED showers in their discussion, I believe that this should not be a reason against publishing the article.
I would therefore like to recommend this paper for publication in SciPost Physics.

---

## Editorial Decision

published